# Classification of Alteration Zones Based on Drill Core Hyperspectral Data Using Semi-Supervised Adversarial Autoencoder: A Case Study in Pulang Porphyry Copper Deposit, China

**Xu Yang** 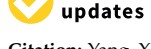, **Jianguo Chen * and Zhijun Chen**

School of Earth Resources, China University of Geosciences, Wuhan 430074, China
* Correspondence: jgchen@cug.edu.cn

**Abstract:** With the development of hyperspectral technology, it has become possible to classify alteration zones using hyperspectral data. Since various altered rocks are comprehensive manifestations of mineral assemblages, their spectra are highly similar, which greatly increases the difficulty of distinguishing among them. In this study, a Semi-Supervised Adversarial Autoencoder (SSAAE) was proposed to classify the alteration zones, using the drill core hyperspectral data collected from the Pulang porphyry copper deposit. The multiscale feature extractor was first integrated into the encoder to fully exploit and mine the latent feature representations of hyperspectral data, which were further transformed into discrete class vectors using a classifier. Second, the decoder reconstructed the original inputs with the latent and class vectors. Third, we imposed a categorical distribution on the discrete class vectors represented in the one-hot form using the adversarial regularization process and incorporated the supervised classification process into the network to better guide the network training using the limited labeled data. The comparison experiments on the synthetic dataset and measured hyperspectral dataset were conducted to quantitatively and qualitatively certify the effect of the proposed method. The results show that the SSAAE outperformed six other methods for classifying alteration zones. Moreover, we further displayed the delineated results of the SSAAE on the cross-section, in which the alteration zones were sensible from a geological point of view and had good spatial consistency with the occurrence of Cu, which further demonstrates that the SSAAE had good applicability for the classification of alteration zones.

**Keywords:** alteration zones; drill cores; measured hyperspectral data; semi-supervised learning; multiscale feature extraction

## 1. Introduction

In porphyry copper deposits, mineralization is closely related to wall rock alteration [1,2], which can be intuitively cataloged and studied by geologists using surface and subsurface geological information. Affected by surface factors such as weathering, the original altered minerals exposed on the surface may change (e.g., oxidation of iron-bearing minerals); therefore, compared with the surface, the alteration extracted from the drill cores can better reflect the distribution of alteration zones associated with the mineralization in the porphyry copper deposit. However, geological logging is a time-consuming, labor-intensive, and qualitative task susceptible to the subjective perception of geologists [3]. With the rapid development of hyperspectral technology, it has become possible to quantitatively and objectively study the alteration minerals using high-resolution spectroscopy from visible and near-infrared (VNIR) to shortwave infrared ranges [4,5]. Since altered rocks are comprehensive representations of relevant minerals mixed in different proportions, for example, Table 1 shows that most of the alterations contain quartz in the porphyry

copper deposit and their spectra are highly similar, which increases the difficulty in classifying them. The current hyperspectral classification methods can be categorized into conventional methods, machine learning, and deep learning methods.

**Table 1.** The altered rocks and typical alteration minerals of different alteration types in porphyry copper deposit.

| Altered Rocks | Alteration Zones | Alteration Types | Typical Alteration Minerals |
|---|---|---|---|
| Quartz monzonite porphyry; Quartz diorite porphyrite; Granodiorite porphyry | Silicified | Silification | Quartz; opal |
| | Potassic | Potassification | Orthoclase; biotite; quartz |
| | Phyllic | Sericitization | Sericite (muscovite/illite); quartz |
| | Argillic | Argillization | Kaoline; montmorillonite; quartz |
| | Propylic | Propylitization | Epidote; chlorite; quartz |
| | Hornfelsic | Horntfels | Hornstone; quartz |

Conventional methods distinguish different rock ores by studying their intrinsic spectral absorption characteristics [6]. The studies by Hamilton et al. [7–9] demonstrated that the structure, general composition, and types of pyroxenes can be determined from the variant trends of absorption shape, depth, and position using the visible and thermal emission spectra. Some feature matching methods, such as the spectral angle map (SAM) [10], spectral information divergences (SID) [11], and spectral feature fitting (SFF) [12], determine the pending spectra compared to a referenced spectral library (e.g., United States Geological Survey (USGS) Spectral Library [13]). To better extract the weak alteration information, Lawrence et al. [14] utilized the reflectance ratio method to enhance the useful information and successfully discriminate the hydrothermally altered rocks using VNIR multispectral images. In addition, some soft classification methods, such as matched filtering (MF) [15], mixture-tuned match filtering (MTMF) [16], and constrained energy minimization (CEM) [17], are adopted to map the alteration domains in an unmixing manner. Although most of the above classification algorithms based on the absorption features have achieved good results in alteration extraction domains, they require a lot of human-computer interactions and subjectivity, which could be difficult to apply to more complex and larger amounts of data.

In recent years, to further improve the accuracy of the classification, machine learning and deep learning have been applied in geological fields [18–20]. These methods can be divided into unsupervised classification, supervised classification, and semi-supervised classification depending on whether the labeled samples are employed for training. Unsupervised methods, such as K-Means [21], ISODATA [22], and GMM [23], mainly perform clustering by calculating the distance or probability between pending spectra and other spectral clusters. Autoencoders (AE) [24] and their variants, such as the adversarial autoencoder (AAE) [25], perform clustering by compressing and reconstructing the original input data in an unsupervised way. Nonetheless, it is also a challenge to determine the type of each cluster from a large number of highly similar spectra. The supervised methods yield the classification results using a classifier obtained by training a batch of labeled samples. For example, Maliheh et al. [26] adopted the support vector machine (SVM) to separate the potassic and phyllic alterations using the fluid inclusion data in the Sungun porphyry copper deposit. However, in actual geological tasks, considering the high cost and efficiency of the work, it is difficult to obtain a large number of training samples; therefore, the semi-supervised classification algorithms, such as the semi-supervised adversarial autoencoder [25] and Self-trained eXtreme Gradient Boosting Trees (SXGBoost) [27], which only require a small number of training samples, have been widely applied to geological fields [28]. In addition, compared with traditional machine learning methods, convolutional networks (CNNs) [29], benefiting from powerful feature extraction capabilities, are widely applied to speech recognition [30,31] and image classification [32,33]. To combine the

advantages of both, semi-supervised CNNs have represented a research trend to further improve classification accuracy [34].

The studies in [35–37] demonstrated that taking full advantage of spectral contexture and multiscale information is beneficial to improving classification accuracy. Moreover, the discrete class vectors represented in the one-hot form should follow a categorical distribution without carrying any style information. Therefore, in the case of available information and limited labeled samples, a novel semi-supervised adversarial autoencoder (SSAAE) was proposed to classify alteration zones using hyperspectral data collected from the drill cores of the Pulang porphyry copper deposit. In the encoder, the CNN-based multiscale feature extractor was first employed to characterize the latent spectral features of hyperspectral data, which were further transformed into discrete class vectors using a classifier. Subsequently, we concatenated the latent continuous spectral features and the discrete class vectors to reconstruct the original spectra in the decoder. Moreover, class vectors were imposed on a categorical distribution using the adversarial process, and the limited labeled samples were employed to guide the training of the network. In view of that, the main research objectives are: (1) to fully characterize measured hyperspectral data related to different alteration zones using the multiscale features; (2) to enhance network performance with limited labeled data; (3) to delineate the alteration zones related to Cu mineralization in the vertical direction and cross-section using the measured hyperspectral data collected from drill cores.

## 2. Geological Settings of Study Area

The Pulang copper deposit is located at the junction of Shangri-La Yunnan Province, Sichuan Province, and the Yidun-Zhongdian Island Arc Edge. As can be seen from Figure 1a, the lithologies in the study area consisting of quartz diorite porphyry, quartz monzonite porphyry, and granodiorite porphyry, were mainly produced by the intense magmatism during the Indonesian island arc stage [38–40].

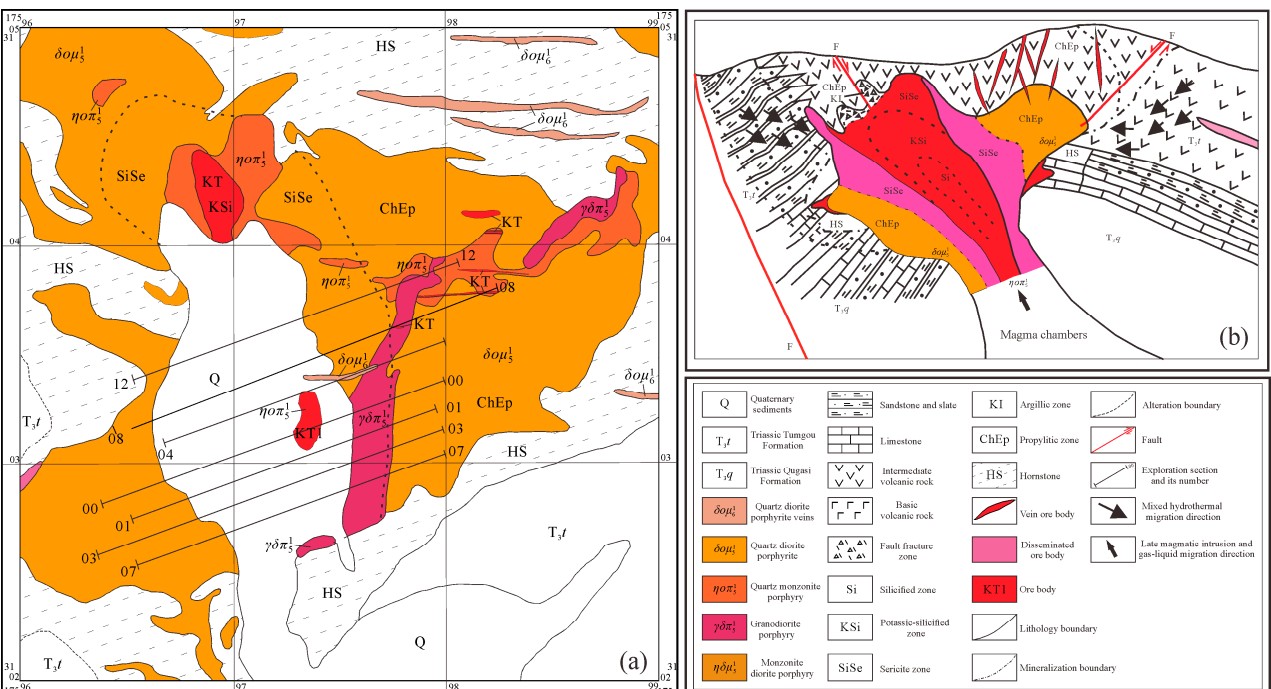

**Figure 1.** (**a**) Geological map of the Pulang mining area (modified from [39]); (**b**) metallogenic model of the Pulang porphyry copper deposit (modified from [41]).

Among them, quartz monzonite porphyry is most closely related to Cu mineralization. Under the late magmatic-hydrothermal action, the rock mass continuously metasomatized with the early protolith, resulting in different periods of medium-low temperature

hydrothermal alteration mineral assemblages, which have alteration zoning features of porphyry copper deposits [42,43].

Since the study area conforms to the porphyry copper deposit model, the alteration from the core to the outer edge can be divided into silicified, potassic, phyllic, prophylitic, and hornfelsic zones, corresponding to alteration types silification, potassification (i.e., potassic feldsparization and biotization), sericitization, propylitization (i.e., chloritization and epidotization), and hornfels (Figure 1b). Affected by multi-stage magmatism, porphyry copper deposits could occur to multiple superpositions of various alterations. Since the argillic zone was locally superimposed on other alterations without an independent existence, this zone was omitted in this study. At present, three areas were delineated as industrial ore bodies, in which KT1 is the main ore body with the widest distribution.

## 3. Data and Method

### 3.1. Hyperspectral Data Collection and Preprocessing

In this research, an Analytical Spectral Devices, Inc. (ASD, Boulder, CO, USA) Field Pro portable spectroradiometer with wavelength coverage from 0.35 to 2.5 μm was employed to efficiently collect the hyperspectral data from the drill cores of the Pulang copper deposit. The spectral resolution of the ASD covers 3–10 nm corresponding to different wavelength ranges, which is sufficient to indicate the absorption features of various alteration minerals. Before the measurement, the instruments were calibrated according to the specifications. The drill cores were evenly split into the fresh plane, which is perceived as the spectral acquisition plane.

The collection of hyperspectral data mainly focused on the obvious alteration and mineralization regions of the drill cores, which can be delineated from geological logs or scanned drill core photographs (Figure 2). Hyperspectral data can be acquired from the specific interval depths (e.g., every 0.4 m) or depth ranges of interest. In this work, seven significant cross-sections were selected around KT1 depicted in Figure 1a, including a total of 31 drill cores with 20,740 hyperspectral data collected; a database was also established to associate the measurement positions with their depths. In addition, preprocessing, including splice correction and denoising, was employed for all measured hyperspectral data, and the limited labeled hyperspectral data for each alteration type were also collected from the region of interest.

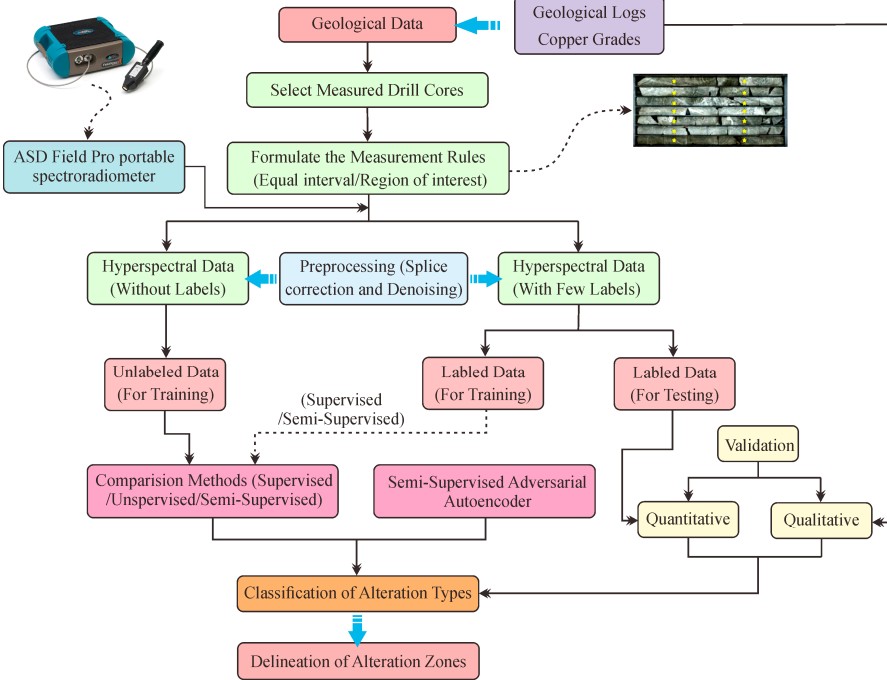

**Figure 2.** An overall flowchart used to delineate the alteration zones.

### 3.2. Method

As the flowchart shown in Figure 2, the SSAAE was proposed to classify the different altered zones using measured hyperspectral data with the schematic diagram depicted in Figure 3. Let $\mathbf{X} \in \mathbb{R}^{N \times B}$ be the original hyperspectral data cube with $N$ data and $B$ bands; $\mathbf{V} \in \mathbb{R}^{N \times M}$ denotes the class vectors with $M$ classes and $\mathbf{Z} \in \mathbb{R}^{N \times 2M}$ is the latent feature vector with a length of $2M$. In the following subsection, the architecture of SSAAE was presented in detail.

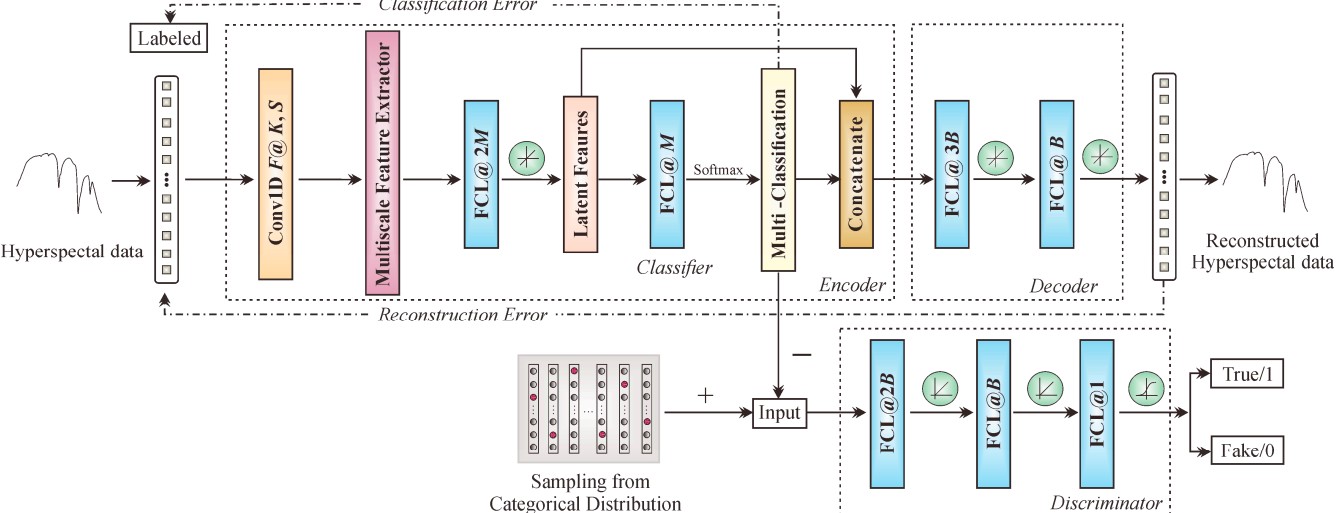

**Figure 3.** The SSAAE architecture for classification of alteration zones using drilling core hyperspectral data. The classifier is trained on datasets given a labeled mini-batch and mostly unlabeled while the multi-classification representation is imposed on a Categorical distribution using an adversarial process. Where Conv1D *F@ K, S* represents a 1D convolutional layer with a filter size of *F*, kernel size of *K*, and stride of *S*, and FCL@ *m* denotes an FCL with the output dimension of *m*.

### 3.2.1. AE of SSAAE

The essential function of AE is to exploit the latent feature representations and classification results of the hyperspectral data using the encoder and to incorporate both to reconstruct the original input using the decoder. Before elaborating on the AE in detail, a multiscale feature extractor was first introduced to fully exploit the multiscale features of the hyperspectral data.

(1)    Multiscale Feature Extractor

The hyperspectral data collected from different alterations may have great similarities regarding absorption features, because each altered rock may contain the same or similar minerals with diverse proportions, which greatly increases the difficulty of distinguishing among them. Since the measured hyperspectral data disregards spatial information, only spectral information can be employed to distinguish different categories. Therefore, to better exploit and mine the characteristics of a single band and its correlation with contextual bands, a CNN-based multiscale feature extractor was integrated into the encoder to extract the multiscale features. As depicted in Figure 4, the multiscale feature extractor comprises the following constituents.

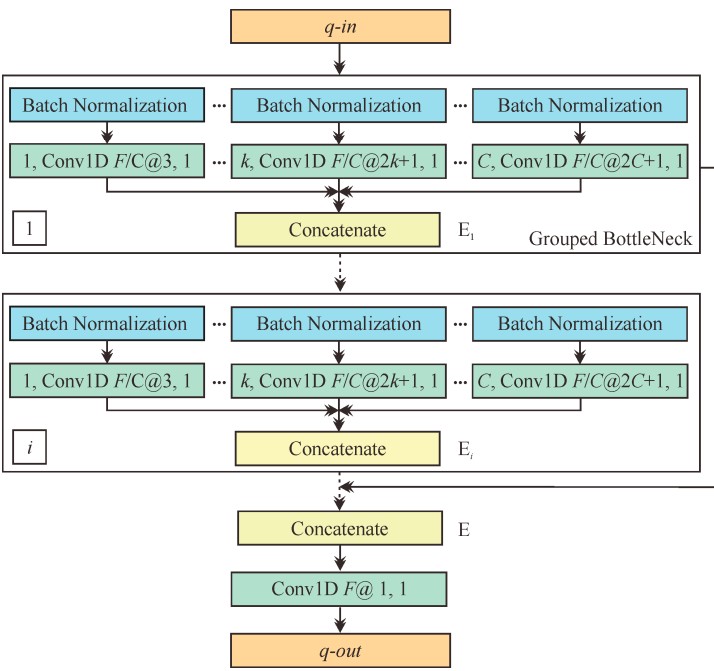

**Figure 4.** The diagram of the multiscale feature extractor. Where $C$ represents the number of bottleneck blocks.

**Grouped bottleneck**: Since the morphological sizes of each absorption characteristic peak of hyperspectral curves are usually different (i.e., different scales), a module with diverse receptive fields is conducive to fully extracting the multi-contextual comprehensive information. Research in [44] reveals that expanding cardinality (i.e., the size of the transformations) is more effective than deepening and widening the network. In this sense, to enhance the characterization ability of the network, we designed a grouped bottleneck combining the idea of inception [45] and grouped convolution [46]. The grouped bottleneck adopted the split-transform-merge [44] strategy implemented using multiple convolutional layers with different perceptive fields to explore different scale features at the same stage. As can be seen from Figure 4, multiscale features, extracted using multiple transformation functions consisting of a Batch Normalization (BN) [47] and a one-dimensional convolutional layer (1D Conv) with different kernel sizes, were merged using a concatenate layer. The process can be defined as follows:

$$E_i = \text{Concat}(E_{i,1}(q), \dots, E_{i,k}(q), \dots, E_{i,C}(q)) \tag{1}$$

where $E_{i,k}(q)$ represents the $k$th transformation function in the $i$th grouped bottleneck $E_i$ with the same input $q$; the transformation function quantities depend on the cardinality denoted as $C$. Since the topology of each grouped bottleneck is the same, the feature extraction capability of the network increases without increasing the parameter complexity, which could well mitigate the detrimental effect of overfitting.

**Global feature fusion:** Each grouped bottleneck at different depths can extract multilevel scale features (i.e., continuous features) and omit the former features extracted from the preceding grouped bottlenecks. To effectively alleviate the direct attenuation of useful feature representations between the indirect connection layers, it is worth integrally fusing the multilevel features to take full advantage of the spectral information and achieve high-dimensional representation. As depicted in Figure 4, the fusion process can be formulated as:

$$E = \mathcal{T}(E_1, \dots, E_i, \dots, E_L) \tag{2}$$

where $\mathcal{T}$ is the fusion function comprising a concatenate layer and 1D Conv with a kernel size of 1; $(E_1, \dots, E_i, \dots, E_L)$ denotes multilevel feature representations extracted using the

grouped bottlenecks at different depths; $L$ represents the number of the grouped bottlenecks. Moreover, the multilevel connection structures can well alleviate the detrimental effects posed by gradient vanishing [48].

(2)   Encoder

The encoder aims to characterize the high-hierarchical feature $\mathbf{Z} \in \mathbb{R}^{N \times 2M}$ from original hyperspectral data $\mathbf{X} \in \mathbb{R}^{N \times B}$ in a dimensionality reduction manner and further convert them into the classification results $\mathbf{V} \in \mathbb{R}^{N \times M}$ using a classier.

As depicted in Figure 3, the first layer, denoted as $O^{(1)}$, is a 1D Conv layer with a filter of $F^{(1)}$, kernel size of $K^{(1)}$, and stride of $S^{(1)}$, which was used to extract the low-hierarchical features by transforming the original input $\mathbf{X} \in \mathbb{R}^{N \times B}$ into higher dimensions. Then the feature maps arranged from the first layer were fed to the second layer $O^{(2)}$, which was used to comprehensively transform the low-hierarchical into high-hierarchical feature presentations of the hyperspectral data using the multiscale feature extractor described above. In the third layer $O^{(3)}$, an FCL with a linear activation was employed to convert the high-hierarchical features into specific-length feature vectors $\mathbf{Z} \in \mathbb{R}^{N \times 2M}$. Thereinto, to ensure that the latent features of the hyperspectral data can be fully represented in low dimensions, we set the length of the feature vectors at twice the classification quantities. The fourth layer $O^{(4)}$, consisting of another FCL connected to a nonlinear softmax activation was perceived as a classifier, which was used to transform the latent feature $\mathbf{Z} \in \mathbb{R}^{N \times 2M}$ into the corresponding class vectors $\mathbf{V} \in \mathbb{R}^{N \times M}$ (i.e., the class vector with a length of $M$ is represented in one-hot form). Last, we concatenated the continuous features and discrete class vectors as the output of the encoder in the fifth layer $O^{(5)}$. The encoder process can be formulated as follows:

$$\mathbf{Y} = \varphi_{E}(\mathbf{X}) \equiv \text{Concat}(\mathbf{Z}, \mathbf{V}) \tag{3}$$

Under the processing of $\varphi_{E}$, the dimension of the original hyperspectral data decreased from $\mathbb{R}^{N \times B}$ to $\mathbb{R}^{N \times 3M}(B >> 3M)$. Additionally, $l_2$ regularizations were applied in the network to mitigate the effects of white noise [49].

(3)   Decoder

The essential function of the decoder is to reconstruct the original hyperspectral data using latent feature vectors $\mathbf{Z} \in \mathbb{R}^{N \times 2M}$ and class vectors $\mathbf{V} \in \mathbb{R}^{N \times M}$, as shown in Figure 3. The decoder comprises two FCLs separately connected to the linear activation function, denoted as $O^{(6)}$ and $O^{(7)}$, which promotes the encoder to characterize the input data using the most relevant instantiation parameters. Considering the input dimension of the $O^{(6)}$ is $\mathbf{Y} \in \mathbb{R}^{N \times 3M}$ and the output dimensions of the decoder are consistent with the original input $\mathbf{X} \in \mathbb{R}^{N \times B}$, we set the output size of $O^{(6)}$ to $3B$ in this network. The formulation of the decoder can be defined by:

$$\hat{\mathbf{X}} = \varphi_D(\mathbf{Y}) \equiv W^{(O^{(7)})}(W^{(O^{(6)})}\mathbf{Y} + b^{(O^{(6)})}) + b^{(O^{(7)})} \tag{4}$$

where $\hat{\mathbf{X}}$ represents the reconstructed hyperspectral data; $W^{(O^{(6)})}$, $b^{(O^{(6)})}$ and $W^{(O^{(7)})}$, $b^{(O^{(7)})}$ denote the weights and biases of the FCLs in layer 6 and layer 7. With the processing of $\varphi_D$, the dimension of the latent codes $\mathbf{Y} \in \mathbb{R}^{N \times 3M}$ is restored to the original input $\mathbf{X} \in \mathbb{R}^{N \times B}$.

### 3.2.2. Adversarial Process of SSAAE

AAE performs variational inference via the max-min adversarial game between the generator $G$ and discriminator $D$ to match the aggregated posterior of hidden vectors to an appropriate prior distribution [25]. Let $q(\mathbf{V})$ be the aggregated posterior distribution of the class vectors $\mathbf{V}$, which can be defined using the hyperspectral data distribution $p_d(\mathbf{X})$ and the encoding function $q(\mathbf{V}|\mathbf{X})$ as follows:

$$q(\mathbf{V}) = \int_{\mathbf{X}} q(\mathbf{V}|\mathbf{X}) p_d(X) \mathrm{d}X \tag{5}$$

For multi-classification problems, the label vectors are usually yielded in one-hot form, which indicates the label vectors comply with the categorical distribution $p(\mathbf{V}) \sim$ Cat $(\mathbf{V})$ without carrying any style information. For this purpose, the regularization of the AAE plays an important role in matching the aggregated posterior $q(\mathbf{V})$ to a categorical distribution $p(\mathbf{V})$.

The generator $G$ of the SSAAE is a branched classification network in the encoder, which requires the aggregated posterior distribution to fool the discriminator $D$ such that $q(\mathbf{V})$ is subject to the prior distribution $p(\mathbf{V})$. The discriminator $D$ consists of 3 FCLs with $2B$, $B$, and 1 unit connected with 2 ReLu and a sigmoid function, respectively. The adversarial process can be divided into two stages using an alternating way: (a) training the discriminator $D$ to distinguish the true samples generated from the categorical distribution and the fake samples (i.e., the multi-classification vectors) provided by the generator $G$; (b) training the generator $G$ to fool the discriminator $D$.

### 3.2.3. Object Loss Function

In this study, the following three loss functions should be considered for classifying the measured hyperspectral data.

(1)    Semi-Supervised Classification Loss

In the semi-supervised classification stage, the class vector $\mathbf{V}$ of a branch output of the encoder was first updated by minimizing the categorical cross-entropy cost on the labeled mini-batches, for which the loss function can be defined as follows:

$$\mathcal{L}_{\mathcal{S}}(\hat{\mathbf{V}}, \mathbf{V}) = -\hat{\mathbf{V}} \log \mathbf{V} \tag{6}$$

where $\hat{\mathbf{V}}$ and $\mathbf{V}$ represent the estimated and true labels with the one-hot form.

(2)    Reconstruction Loss

In the reconstruction stage, SSAAE reconstructed the original input spectra by continuously updating $\hat{X}$, which should be considered in a twofold way: the reflectance and shape similarity, respectively. The reflectance diversity between the reconstructed and original spectra can be assessed using the mean absolute error (MAE), which has a glaring deficiency due to its sensitivity to spectral variability and noise. The spectral shape similarity can be evaluated using the spectral angle distance (SAD), which is scale-invariant and insensitive to noise; whereas it does not consider the magnitude of spectral reflectance. In this sense, to better reconstruct the hyperspectral spectra, we comprehensively considered the reflectance and shape similarity when defining the reconstruction error (RE) to mitigate detrimental effects such as spectral variability and noise. The RE function can be formulated by

$$\begin{aligned}
\mathcal{L}_{\text{RE}}(\mathbf{X}, \hat{\mathbf{X}}) &= \mathcal{L}_{\text{SAD}}(\mathbf{X}, \hat{\mathbf{X}}) + \mathcal{L}_{\text{MAE}}(\mathbf{X}, \hat{\mathbf{X}}) \\
\mathcal{L}_{\text{SAD}}(\mathbf{X}, \hat{\mathbf{X}}) &= \cos^{-1}\left(\frac{\mathbf{x}^{\mathsf{T}} \hat{\mathbf{x}}}{||\mathbf{x}||_2 ||\hat{\mathbf{x}}||_2}\right) \\
\mathcal{L}_{\text{MAE}}(\mathbf{X}, \hat{\mathbf{X}}) &= ||\mathbf{X} - \hat{\mathbf{X}}||_1
\end{aligned} \tag{7}$$

In addition, since each band was updated independently during training, to better promote the smoothness of the reconstructed spectra, the weight update amplitudes of the two FCLs were constrained in the decoder using a total variation (TV) regularization [50]. The regularization can be formulated as follows:

$$\begin{aligned}
\mathcal{R}_{sm(\text{O}^{(6)})} &= \lambda_1 \sum_{i=1}^{3M} \sum_{j=1}^{3B-1} \left|\left| W_{i,j+1}^{(\text{O}^{(6)})} - W_{i,j}^{(\text{O}^{(6)})} \right|\right|_2 \\
\mathcal{R}_{sm(\text{O}^{(7)})} &= \lambda_1 \sum_{i=1}^{3B} \sum_{j=1}^{B-1} \left|\left| W_{i,j+1}^{(\text{O}^{(7)})} - W_{i,j}^{(\text{O}^{(7)})} \right|\right|_2
\end{aligned} \tag{8}$$

(3)    Adversarial Loss

In the adversarial regularization process, the discriminator $D$ was first updated by distinguishing the true samples generated from the categorical distribution and the fake samples provided from the multi-classification result, a process that can be formulated as follows:

$$\mathcal{L}_D = \lambda_2 \sum_{i=1}^{N} \left( \hbar_i \log D(\mathbf{V}_i^+) + (1 - \hbar_i) log(1 - D(\mathbf{V}_i^-)) \right) \tag{9}$$

The generator was then updated by confusing the discriminator $D$; that is, the discriminator $D$ was fooled into perceiving the generator output as the positive samples (i.e., true samples), which the formulation can be defined by:

$$\mathcal{L}_G = \lambda_2 \sum_{i=1}^{N} \hbar_i \log D(\mathbf{V}_i^-) \tag{10}$$

where $\mathbf{V}_i^+$ and $\mathbf{V}_i^-$ represent the $i$th positive sample generated using the categorical distribution and the negative sample provided from the output of the generator; $\hbar_i$ denotes that the sample is true (i.e., 1) or false (i.e., 0), which is perceived as the label for the discriminator $D$.

## 4. Experimental Results and Analysis

In this section, the comparison experiments on synthetic and measured hyperspectral datasets were performed to certify the validity of the SSAAE in the classification of alteration zones.

### 4.1. Experimental Setup

To evaluate the performance of the SSAAE, we compared it with unsupervised, supervised, and semi-supervised classification methods, which are listed in Table 2. In the case of SVM, the kernel function was set to radial basis function (RBF) and the optimal hyperplane parameter that controls the amount of data involved in the hyperplane and its margins computation was determined to be 10. For SSAAE, Adam [51] was employed as the optimizer. Furthermore, to better ensure the fairness of the comparative experiments, the training tolerances for K-Means, GMM, and SVM were set to $1 \times 10^{-3}$, and all training epochs (i.e., maximum iterations) were determined to 200 and repeated five times independently to avoid the errors posed by the random initialization process.

**Table 2.** The classification methods for the comparison experiment.

| Method | Description | Unsupervised | Supervised | Semi-Supervised |
|---|---|:---:|:---:|:---:|
| K-Means [21] | Calculating the distance between pending spectra and cluster centroids | ● | ○ | ○ |
| GMM [23] | Computing the likelihood of pending spectra in each certain Gaussian distribution | ● | ○ | ○ |
| SAM [10] | Calculating the spectral angle distance between the pending and reference spectra | ○ | ● | ○ |
| SVM [26] | Finding the margin-maximizing hyperplane in the feature space | ○ | ● | ○ |
| SXGBoost [27] | Self-trained eXtreme Gradient Boosting Trees | ○ | ○ | ● |
| AcPCKMeans [52] | Active Semi-Supervision for Pairwise Constrained K-Means | ○ | ○ | ● |

● Yes ○ No.

In the comparison experiment, the confusion matrices, overall accuracy (OA), average accuracy (AA), and Kappa [53] were used to quantitatively evaluate the classification

accuracy. For the measured hyperspectral dataset of the drill cores from the Pulang copper deposit, we also assessed the classification results from a geological view qualitatively.

### 4.2. Experiment with Synthetic Dataset

#### 4.2.1. Dataset Description

The synthetic dataset was employed to simulate the spatial distribution of the lithologies or altered rocks in the drill cores. To ensure that the synthetic drill cores are more reasonable, five mineral spectra (i.e., montmorillonite, chlorite, epidote, muscovite, and quartz) relevant with alterations were selected from USGS spectral library as the endmembers, depicted in Figure 5a, and their corresponding abundance maps were yielded using the Gaussian field method [54]. The synthetic dataset was generated using the linear mixing model [55], which is perceived as 100 drill cores (in rows) with a depth of 100 m (in the column); that is, each pixel is isolated with a sampling interval of 1 m. For visualization purposes, we presented the mixed hyperspectral dataset as shown in Figure 5b. In addition, in the experiment, 20 labeled data for each class were selected to assess the effectiveness of the network with few labeled data.

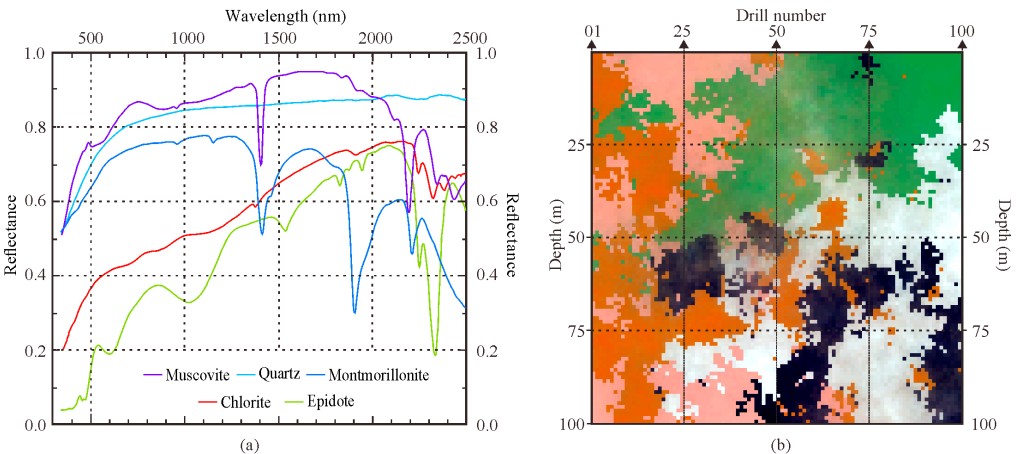

**Figure 5.** (**a**) The spectra selected from USGS; (**b**) Synthetic dataset with fake color (1040 nm, 2340 nm, and 1415 nm). Where row and column present the drill core number and the depth, respectively.

#### 4.2.2. Hyperparameter Settings

In this section, we explored the optimal hyperparameters of the SSAAE by ablation studies on the synthetic dataset. The classification results, consisting of OA, AA, and Kappa, were obtained with the network trained 200 times and repeated five times independently.

**Impact of Cardinality (*C*) and grouped bottlenecks (*L*)**: Figure 6 illustrates the effect of different numbers of cardinalities and bottlenecks on classification accuracy. For cardinality, the result shows that with the increase of cardinality, the classification accuracy also increased to a certain extent, indicating that the convolution kernels with diverse perceptual fields can better capture multiscale spectral features, thereby improving the classification accuracy. When the cardinality exceeded four, the results were not affected dramatically; therefore, we set the optimal cardinality as four on the premise of fewer training parameters. For bottlenecks, the results indicate multiple bottlenecks were conducive to extracting the multilevel features, thereby enhancing the performance of the network. When the number of bottlenecks exceeded two, the network was slightly degraded. Therefore, we set the optimal value for the number of bottlenecks as two.

**Impact of the adversarial process ($lr_{adv}$)**: The effect of different learning rates for adversarial training ($lr_{adv}$) is depicted in Figure 7a, in which the results illustrate that imposing the categorical distribution on the class vectors is beneficial to enhancing the interpretability of class representations and improving the accuracy of classification. When the $lr_{adv}$ was set to $5 \times 10^{-5}$, the network yielded better performance, indicating that the classification result can be fine-tuned with an appropriate prior distribution.

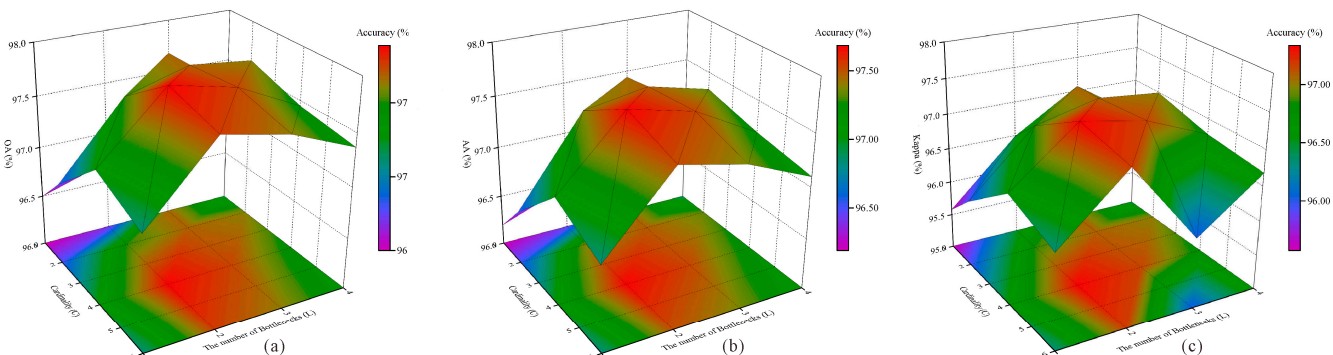

**Figure 6.** The performance of SSAAE on different numbers of cardinalities and bottlenecks. (**a**) OA; (**b**) AA; (**c**) Kappa.

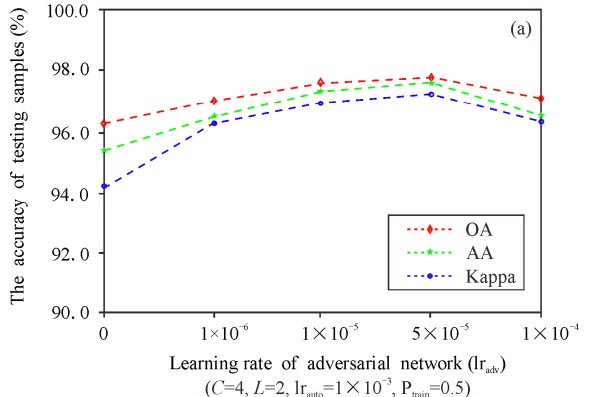 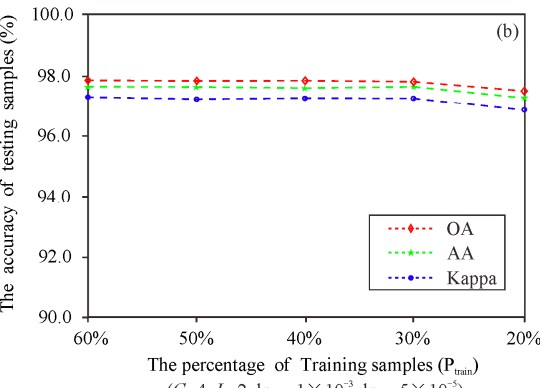

**Figure 7.** The performance of SSAAE with different learning rates of adversarial network and training samples, where $lr_{Adv} = 0$ represents without the adversarial process. (**a**) Impact of adversarial network on classification accuracy; (**b**) impact of the percentage of training samples on classification accuracy.

In addition, the impact of the percentage of training samples ($P_{train}$) is also exhibited in Figure 7b, in which the results prove that when the percentage of the training set was more than 30% (i.e., 30–60%), the performance of the classification result was insensitive to the proportions of the training set, indicating that the SSAAE has better robustness with fewer training samples.

### 4.2.3. Performance Comparison

The classification results of all comparison methods are listed in Table 3, where we can observe that K-Means and SVM, as the classic spectral classification algorithms, achieved better results than GMM. AcPCKMeans, a variant of K-Means, embedded the idea of active learning and pairwise constrained clustering into K-Means to achieve semi-supervised clustering. Compared with K-Means, the classification accuracy of AcPCKMeans on this dataset was slightly enhanced. SAM was difficult to distinguish the local features of the mixed spectra with high similarity. Regarding the high similarity and complexity of the mixed hyperspectral data, SXGBoost did not obtain good classification results on training sets with limited labels.

Conversely, SSAAE outperformed other methods in terms of classification accuracy with limited labels. For illustrative purposes, Figure 8a,b show the distribution of original spectra and latent vectors, in which the result proves that SSAAE benefiting from the multi-scale feature extraction module, can better separate feature space. In addition, considering that the convergence process of the reconstruction error on the training and testing sets can well reflect the performance of the optimization, the results in Figure 8c illustrate the convergence speed of the network was fast while the training and testing processes were consistent, indicating that the network was not over-fitting and was well optimized.

**Table 3.** Comparison of classification results using different methods on the synthetic dataset.

| Class | Unsupervised | | Supervised | | Semi-Supervised | | | Reference |
| --- | --- | --- | --- | --- | --- | --- | --- | --- |
| | K-Means | GMM | SAM | SVM | SXGBoost | AcPCKMeans | SSAAE | |
| Montmorillonite | **2019** | **2019** | 1859 | **2019** | 1782 | **2019** | **2019** | 2034 |
| Chlorite | 2490 | 2490 | 2468 | **2496** | 2422 | 2490 | 2477 | 2520 |
| Epidote | 1319 | 1341 | 1051 | 1341 | 1092 | 1325 | **1412** | 1479 |
| Muscovite | 1538 | 1320 | **1540** | **1540** | 1368 | 1540 | **1540** | 1574 |
| Quartz | 2349 | 2349 | **2354** | 2344 | 2130 | 2349 | 2341 | 2393 |
| OA(%) | 97.15 | 95.19 | 92.72 | 97.40 | 87.94 | 97.23 | **97.89** | 100.00 |
| AA(%) | 96.56 | 94.13 | 91.23 | 96.89 | 86.64 | 96.67 | **97.68** | 100.00 |
| Kappa(%) | 96.39 | 93.90 | 90.76 | 96.71 | 84.66 | 96.49 | **97.33** | 100.00 |

where the best result is indicated in bold.

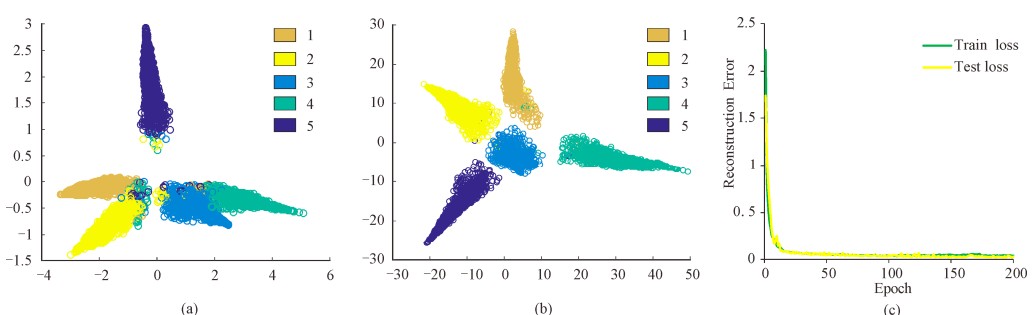

(a)                                      (b)                                      (c)

**Figure 8.** (**a**) The distribution of the original synthetic dataset; (**b**) the distribution of the latent vectors extracted from SSAAE; (**c**) the convergence process of reconstruction error on training and testing sets. Where the dimensionalities were reduced using PCA [56] and the different colors in (**a**,**b**) represent different categories. 1—Montmorillonite; 2—Chlorite; 3—Epidote; 4—Muscovite; 5—Quartz.

In addition, we also established the relationship between latent features and spectral absorption features to indirectly illustrate the interpretability of latent features. Figure 9a shows the spectral absorption features of an epidotization alteration at the wavelength range of 2100–2430 nm, including absorption position, absorption depth, absorption area, and absorption symmetry, which the calculation steps can refer to [57]. Figure 9b–d illustrate that the latent feature representations of 15 epidotization alterations have a strong correlation with the spectral absorption index (i.e., absorption depth and the 6th latent feature, absorption area and the 10th latent feature, and absorption symmetry and the 8th latent feature), which indicates there is good correspondence between them, indirectly verifying the interpretability of the latent features from the spectral level.

### 4.2.4. Robustness to Noise

In this section, SNRs of 10, 20, 30, and 40 dB were added to the synthetic dataset to evaluate the robustness of all methods to diverse noises. According to the hyperparameter sensitivity experiments described above, we set $C$, $L$, and $lr_{adv}$ as 4, 2, and $5 \times 10^{-5}$, respectively. $F$, $K$, $S$, $lr_{auto}$, batchsize, $\lambda_1$, $\lambda_2$, were set to 32, 15, 9, $1 \times 10^{-3}$, 256, $1 \times 10^{-3}$, and $1 \times 10^{-1}$. The comparison results shown in Figure 10 indicate that supervised classification methods were more susceptible to noises than unsupervised methods; that is, K-Means and GMM were more robust than SAM and SVM under diverse SNRs. Furthermore, we can observe that SVM, SXGBoost, and AcPCKMeans were more sensitive to diverse noises under the premise of limited labeled data. In contrast, SSAAE, as a semi-supervised classification method, outperformed other methods in terms of classification accuracy with diverse SNRs, which could be conducive to the $l_2$ regularization used in the network. For illustration purposes, Figure 11 intuitively presents the classification results of all methods under the synthetic dataset with an SNR of 20 dB. The result shows that SSAAE was superior to other methods in terms of classification accuracy, which is consistent with the results illustrated in Figure 10.

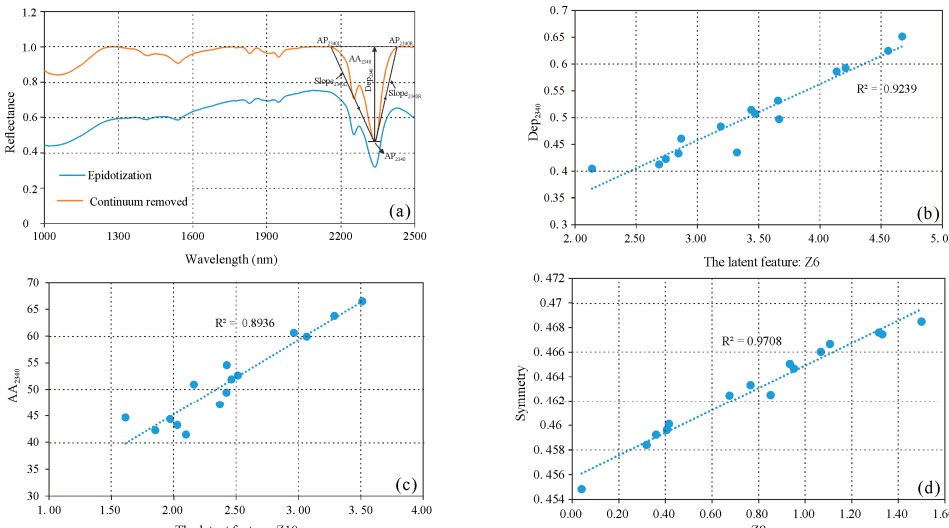

**Figure 9.** Maps of spectral absorption features and their correspondence to latent features. (**a**) The absorption features of the 40th mixed data at the wavelength range of 2100–2430 nm; (**b**) the correspondence between absorption depth (Dep2340) and the 6th latent feature (Z6); (**c**) the correspondence between absorption area (AA2340) and the 10th latent feature (Z10); (**d**) the correspondence between absorption symmetry and the 8th latent feature (Z8).

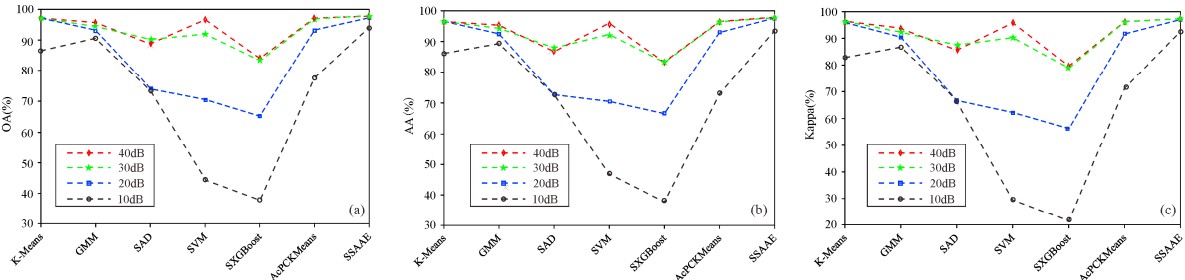

**Figure 10.** Comparison of classification results using different methods under the synthetic dataset with diverse SNRs. (**a**) OA; (**b**) AA; (**c**) Kappa.

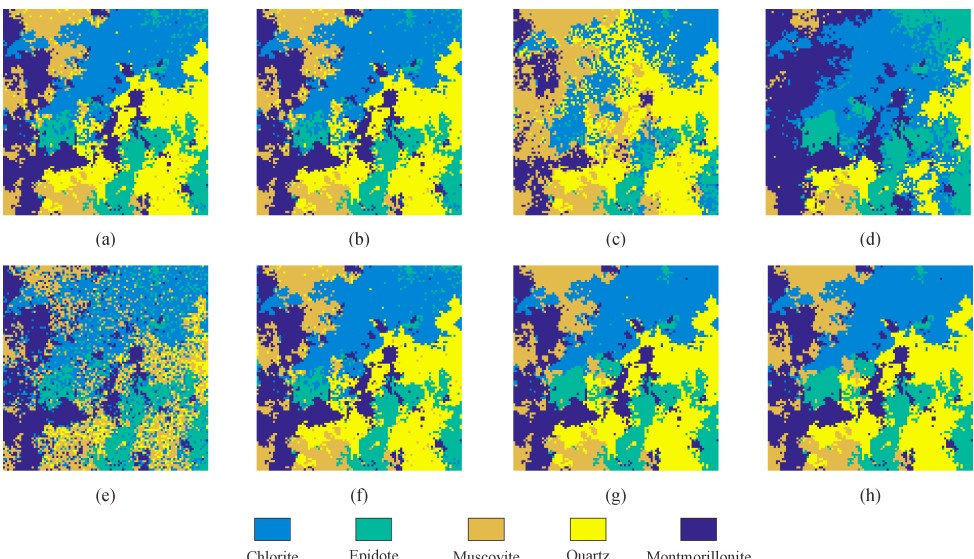

**Figure 11.** Comparison of classification results using different methods under the synthetic dataset with an SNR of 20 dB. (**a**) K-Means; (**b**) GMM; (**c**) SAM; (**d**) SVM; (**e**) SXGBoost; (**f**) AcPCKMeans; (**g**) SSAAE; (**h**) Reference.

### 4.3. Experiment with Drilling Core Hyperspectral Dataset

In porphyry copper deposits, since the alteration types are mainly related to mineralization, we focused on the classification of all alteration zones and the major lithology, including silicified, potassic, phyllic, propylic (i.e., epidotization and chloritization), hornfelsic, and quartz monzonite porphyry. Other unidentified spectra were also added to the classification categories. In addition, to further collect the prior knowledge about the spectral features of the alteration minerals, we also collected the referred spectra from the positions where various alteration types were intensively developed (Figure 12). Moreover, each alteration zone was accompanied by limited labeled data measured from the drill core region of interest, with their quantities shown in Table 4.

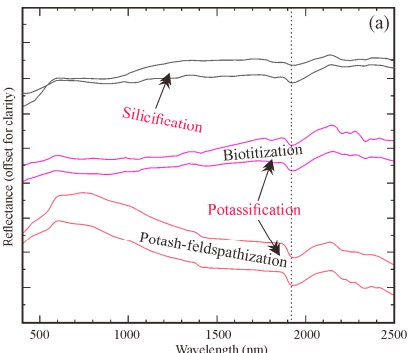 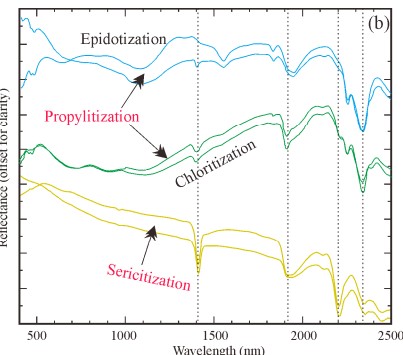

**Figure 12.** Sample spectra of alteration minerals measured from drill cores of the Pulang copper deposit. (**a**) Silification and potassification; (**b**) sericitization and propylitization. These sample spectra were only selected as representative of the alteration, and many drill core spectra have some subtle differences.

**Table 4.** The number of labeled samples on the measured hyperspectral dataset.

| Class No. | Class | Training Samples | Testing Samples |
|:---:|:---:|:---:|:---:|
| 1 | Quartz monzonite porphyry | 54 | 36 |
| 2 | Silification | 64 | 46 |
| 3 | Potassification | 44 | 34 |
| 4 | Sericitization | 89 | 57 |
| 5 | Epidotization | 53 | 39 |
| 6 | Chloritization | 79 | 51 |
| 7 | Hornfels | 61 | 40 |
| 8 | Others | 48 | 31 |
| | Total | 492 | 334 |

Given the high correlation between the measured hyperspectral data, we set the number of grouped bottlenecks $L$ and classifications $M$ to 3 and 8, respectively. Other hyperparameters were the same as in the previous experiment. Moreover, we performed quantitative and qualitative assessments of the classification results using the labeled samples, geological logs, and Cu grades of the drill cores.

#### 4.3.1. Quantitative Evaluation

Table 5 lists the comparative results classified using seven methods. The results indicate that the supervised methods were greater than that of unsupervised methods, where K-Means yielded a better result than GMM in the measured hyperspectral dataset. SAM, SVM, and SXGBoost achieved limited classification accuracy on this dataset with the limited labeled data, which could be related to their difficulty in capturing subtle features from the highly correlated hyperspectral data. The classification accuracy of AcPCKMeans was similar to that of K-Means (even slightly lower than K-Means) on a more complicated hyperspectral dataset. Benefiting from the multiscale feature extractor,

the SSAAE can capture more subtle single-band and correlation with its contextual bands from the measured hyperspectral data; therefore, its classification accuracy was superior to other representative methods, which can be reflected from the comparison results of OA, AA, and Kappa in Table 5.

**Table 5.** Comparison of alteration type classification results using different methods on the measured hyperspectral dataset.

| Class | Unsupervised | | Supervised | | Semi-Supervised | | | Reference |
|---|---|---|---|---|---|---|---|---|
| | K-Means | GMM | SAM | SVM | SXGBoost | AcPCKMeans | SSAAE | |
| Quartz monzonite porphyry | **20** | 15 | 17 | 19 | 7 | 11 | 18 | 36 |
| Silification | 33 | 29 | 28 | 30 | 26 | 35 | **46** | 46 |
| Potassification | **25** | 16 | 15 | 18 | 8 | 21 | 21 | 34 |
| Sericitization | 43 | 45 | 44 | 45 | 44 | 46 | **52** | 57 |
| Epidotization | 27 | 22 | 17 | 20 | 10 | 26 | **38** | 39 |
| Chloritization | 37 | 30 | 31 | 34 | 31 | **39** | 35 | 51 |
| Hornfels | 34 | 31 | 31 | 34 | 31 | 30 | **40** | 40 |
| Others | 18 | 17 | 12 | 14 | 12 | **26** | 25 | 31 |
| OA(%) | 70.96 | 61.38 | 58.38 | 64.07 | 50.60 | 70.06 | **82.34** | 100.00 |
| AA(%) | 69.38 | 59.79 | 56.25 | 61.53 | 47.42 | 68.89 | **81.21** | 100.00 |
| Kappa(%) | 66.41 | 55.41 | 51.92 | 58.40 | 42.69 | 65.44 | **79.65** | 100.00 |

where the best result is indicated in bold.

In addition, we selected two drill cores (i.e., ZK0113, and ZK0307) with well-developed alteration types, detailed geological logs, and available Cu grades to qualitatively discuss the classification effects of all methods.

### 4.3.2. Qualitative Evaluation with ZK0113 and ZK0307

ZK0113 was drilled from the surface with a total of 224.65 m, in which the upper 40 m were Quaternary residual and alluvial deposits. This drill core mainly developed light gray and light gray-brown quartz monzonitic porphyry with porphyry texture and massive structure. In this section, we focused on the alteration types zoned using seven methods in the depth range of 40–220 m, and qualitatively assessed the results using geological logs and the trend of Cu grade, in which the mineralization of Cu is mainly related to the potassic zone and followed by the sericite zone [39].

Affected by multi-stage magmatism, porphyry copper deposits could occur in multiple superpositions of different alteration zones; therefore, the alteration types, including silicification, potassification, sericitization, and propylitization (i.e., chloritization and epidotization), were developed in the depth ranges. We can observe from the geological log in Figure 13a that the alteration was dominated by the phyllic zone. Except for SAM, other methods were generally sensible from a geological point of view. However, in the local vertical ranges, the distributions of the alteration were slightly different. According to the geological log and the trend of Cu grades, the chloritization was locally developed in the depth range of 30.0–48.9 m (①) and 167.2–187.3 m (⑦), especially in the depth range of 107.7–126.0 m (④), which the chloritization was intensively developed, corresponding to a lower Cu grade (< 0.2%). Compared with other methods, SSAAE can better capture the local chloritization while consistent with the trend of Cu grade in these ranges. In addition, the potassic zone corresponding to higher Cu grades (>0.3%) develops well in the depth ranges of 91.8–107.7 m (③) and 126.0–142.8 m (⑤), in which SAM and SSAAE successfully capture the alteration, but SAM achieved a poor classification result in other local ranges. In general, SSAAE outperformed other methods in the classification results of alteration types from global to local in this drill core. Moreover, SSAAE yielded a more detailed classification result than manual geological logs while more consistent with the trend of Cu grades.

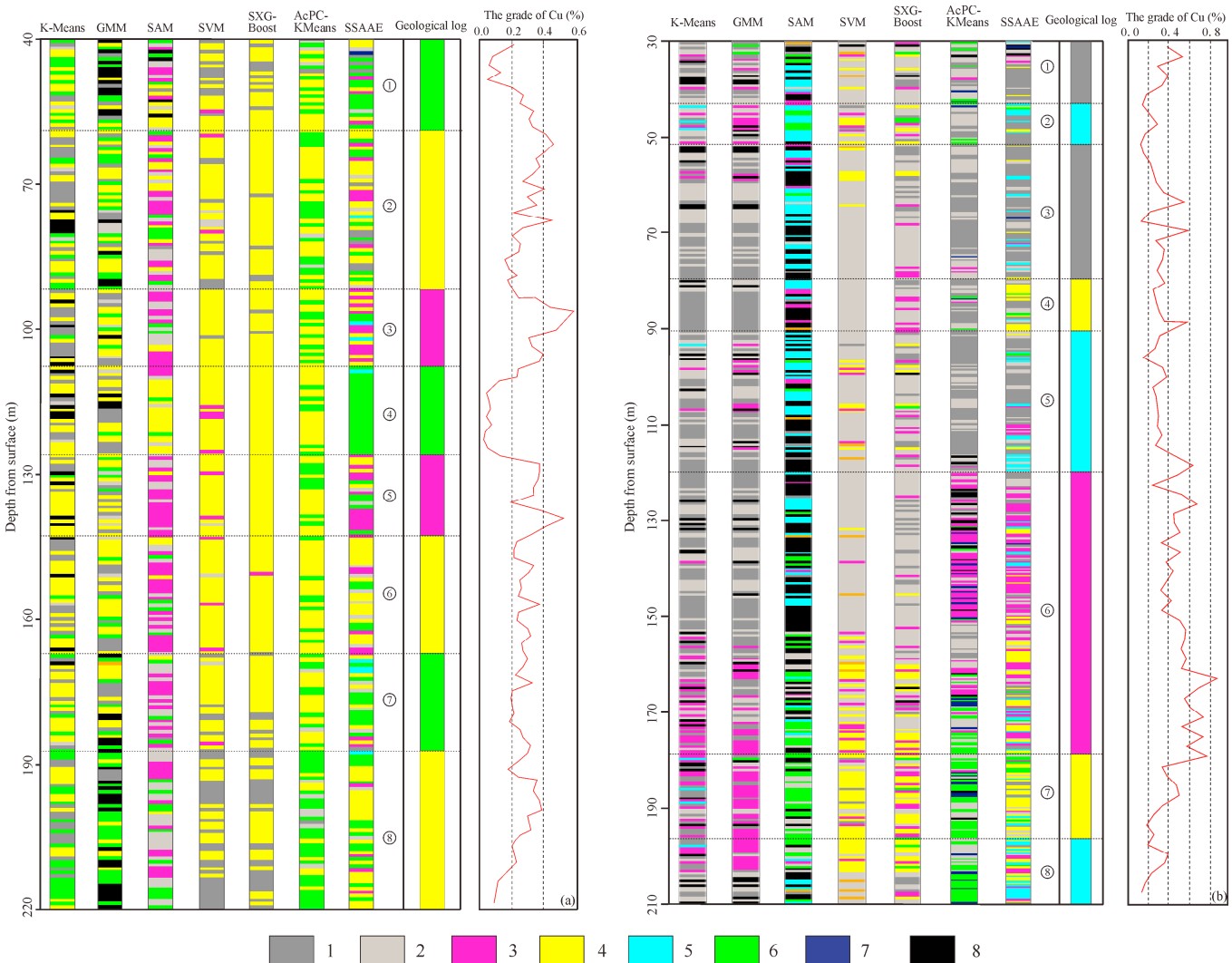

**Figure 13.** Comparison of classification results using different methods under the measured hyperspectral dataset of drilling cores of Pulang copper mine. (**a**) ZK0113; (**b**) ZK0307. 1—Quartz monzonite porphyry; 2—Silification; 3—Potassification; 4—Sericitization; 5—Epidotization; 6—Chloritization; 7—Hornfels; 8—Others.

ZK0307 was cored from the surface to 225.22 m, in which the lithology in the range of 0–30 m were Quaternary residual and alluvial deposits. The lithology of this borehole was mainly light gray quartz monzonite porphyry with porphyry texture and a massive structure. The main alteration types of this drill core were silicification, propylitization, and potassification. In this section, we selected the depth range of 30–210 m as the region of interest.

The geological log in Figure 13b shows that this drill core in the depth range of 30.0–90.5 m (①–④) was dominated by quartz monzonite porphyry. Moreover, according to the trend of Cu grades, the rocks in this range may develop weak alteration, which could be difficult to observe by a manual geological log. In the depth range of 90.5–120.0 m (⑤), comprehensively considering the geological log and trend of Cu grades (close to 0.3%), the development of epidotization may be relatively weak. In the depth range of 120.0–210.0 m (⑥–⑧), the alteration type was dominated by potassification, corresponding to relatively higher Cu grades (>0.4%). According to Figure 13b, SAM, SVM, and AcPCK-Means failed to capture the weak alteration types. On the contrary, K-Means, GMM, and SSAAE were generally consistent with the geological log, but SSAAE could better capture the locally weak epidotization in the depth range of 43.0–51.5 m (②), 90.5–120.0 m

(⑤), and 196.2–210.0 m (⑧), the sericitization in the depth range of 79.6–90.5 m (④) and 178.7–196.2 m (⑦); meanwhile, the alteration zones were in agreement with the trend of Cu grades.

As described above, the alteration zones classified using the unsupervised methods K-Means and GMM and the semi-supervised method SSAAE were generally consistent with the geological logs, while SSAAE can better capture subtle differences between diverse alteration minerals; therefore, the alteration zoning was more accurate, which may benefit from the multiscale feature extractor. In general, SSAAE was superior to other representative methods regarding the classification of alteration zones on the synthetic and measured hyperspectral datasets. In addition, it can also be seen that the classification of alteration zones using hyperspectral data can capture the superposition information of local alteration more objectively and in more detail, which could be difficult to observe with manual geological logging.

### 4.3.3. Qualitative Evaluation with No. 00 Exploration Cross-Section

To further demonstrate the rationality and accuracy of the SSAAE, we also compared the alteration zones with the Cu grades using six drill cores on the No. 00 exploration cross-section. As described below, the Cu grades of the main ore body (i.e., KT1) are continuous and high and gradually decrease to the surrounding area with branching occurring [58]. Figure 14 exhibits the corresponding relationship between the alteration zones delineated using the SSAAE and the Cu grades on the cross-section.

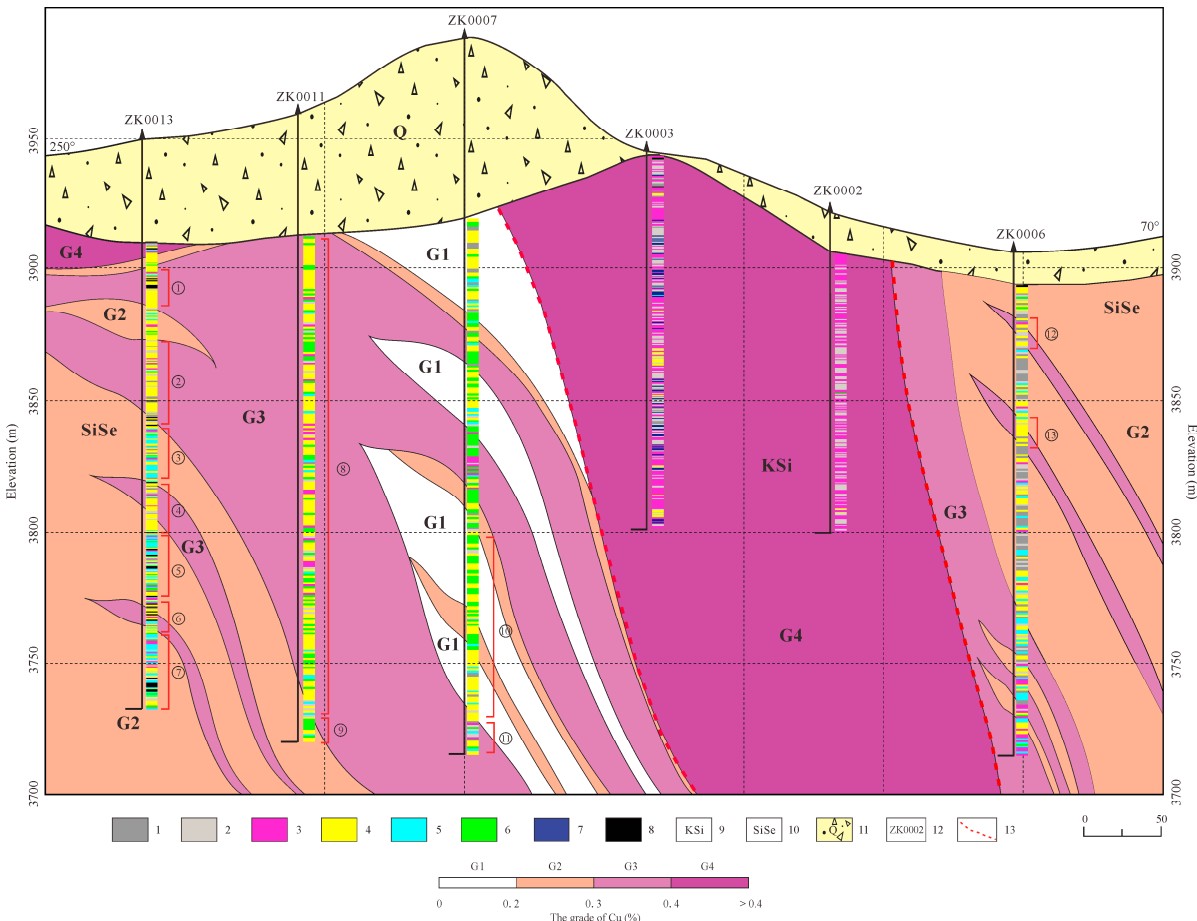

**Figure 14.** Schematic diagram of the corresponding relationship between the zoning result using the SSAAE and Cu grade in the No. 00 exploration cross-section. 1—Quartz monzonite porphyry; 2—Silification; 3—Potassification; 4—Sericitization; 5—Epidotization; 6—Chloritization; 7—Hornfels; 8—others; 9—potassic-silicified zone; 10—Sericite zone; 11—Quaternary; 12—Drill core and its number; 13—Boundary of alteration zones.

As can be seen from Figure 14, we can observe that the alteration types of the ZK0002 and ZK0003 were dominated by potassification and silicification corresponding to the potassium-silicification zone, in which the Cu grade is more than 0.4%, corresponding to the core part of KT1 (i.e., G4 area). The Cu grades on both sides gradually decreased as branching occurred. Among them, from ZK0013 in the depth of 50–62 m (①) and 76–106 m (②), 130–148 m (④) and 172–185 m (⑥), ZK0011 in the depth of 90–210 m (⑧), ZK0007 in the depth of 250–278 m (⑪), and ZK0006 in the depth of 130–190 m (⑫) and 62–74 m (⑬), we can observe when the sericitization was strongly developed or alternated with the potassification, the Cu grade mainly ranged from 0.3% to 0.4% (i.e., G3 area). According to the classification results on ZK0013 in the depth of 109–130 m (③), 148–172 m (⑤), 185–210 m (⑦), ZK0011 in the depth of 210–245 m (⑨), and ZK0007 in the depth of 186–250 m (⑩), sericitization generally developed or alternated with the propylitization in the local area, the Cu grade corresponds to 0.2–0.3% (i.e., G2 area) while the propylitization was strongly developed, the Cu grade is lower than 0.2% (i.e., G1 area). The superposition of phyllic and prophylitic zones increases the difficulty of zoning. Considering that the Cu grade can also be acceptable (>0.2%) when the phyllic and prophylitic zone were superimposed, these zones were classified as phyllic zones.

In summary, the potassic zone corresponds to strong mineralization of Cu, followed by the phyllic zone, while the prophylitic zone corresponds to poorer Cu mineralization. Furthermore, the alteration zones were symmetrically distributed, centered on the KT1; that is, the potassic and silicified zones were located in the center, and the phyllic zone was distributed in the periphery of the potassic zone. The analysis results were consistent with the existing geological data [39], which also qualitatively demonstrate the rationality and accuracy of the SSAAE for the classification of alteration zones using measured hyperspectral data of drill cores.

## 5. Conclusions

In this study, the SSAAE was proposed to classify the alteration zones using the measured hyperspectral data. Regarding the high spectral similarity between altered rocks, in the encoder, a CNN-based multiscale feature extractor was used to fully exploit and mine the multiscale and multilevel continuous latent features of the hyperspectral data and further transform them into discrete classes to better distinguish the alteration types. The decoder used the continuous latent feature and the discrete class vector to reconstruct the original inputs, while the discrete class vectors represented in the one-hot form were matched to a category distribution using an adversarial regularization process. In addition, to fully use the limited labeled samples, a supervised classification process was incorporated into the SSAAE to better guide the training of the network. We also used the TV to smooth the reconstruction data and $l_2$ regularization to mitigate the effects of white noise in the network. The SSAAE and six other representative methods were compared and discussed on the synthetic dataset and the measured hyperspectral dataset of the Pulang copper deposit in quantitative and qualitative ways. The results indicate that SSAAE outperformed six other methods in the rationality and accuracy of alteration zones. Moreover, the classification results in the cross-section further demonstrate that SSAAE had good applicability to classifying the alteration zones. Overall, the proposed method can provide a rapid, objective, and accurate interpretation for the classification of alteration zones using the drill core hyperspectral data in the Pulang copper deposit.

**Author Contributions:** Conceptualization, X.Y. and J.C.; methodology, X.Y. and J.C.; software, X.Y.; validation, X.Y. and Z.C.; formal analysis, J.C.; investigation, X.Y. and Z.C.; data curation, Z.C.; writing—review and editing, X.Y.; supervision, J.C. and Z.C.; funding acquisition, J.C. and Z.C. All authors have read and agreed to the published version of the manuscript.

**Funding:** This work was funded by the National Key R&D Program of China (Grant No. 2022YFF0801201, 2017YFC0601500, and 2017YFC0601504) and the Key enterprise project of Yunnan Diqing Nonferrous Metals Co., Ltd., Shangri-la (Grant No. 2018026251).

**Data Availability Statement:** The data presented in this study are available on request from the corresponding author.

**Acknowledgments:** Thanks to the leaders of the Yunnan Diqing Nonferrous Metals Co., Ltd., Shangri-la for providing access to the drill cores and relevant geological data. Yuchen Zhu, Fucheng Zhang, Xiaohu Guo, and Rui Liu are acknowledged for hyperspectral data acquisition. We also appreciate A.H. Dijkstra and W.H. Bakker from the Faculty of Geo-Information Science and Earth Observation (ITC), University of Twente, and the anonymous reviewers for their valuable suggestions.

**Conflicts of Interest:** The authors declare no conflict of interest.

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
