# Peer review of "Classification of Alteration Zones Based on Drill Core Hyperspectral Data Using Semi-Supervised Adversarial Autoencoder: A Case Study in Pulang Porphyry Copper Deposit, China"

_remotesensing, doi:10.3390/rs15041059_

Round 1

Reviewer 1 Report

This manuscript proposes a semi-supervised adversarial autoencoder(SSAAE) to classify the alteration zones using the drill core hyperspectral data collected from the Pulang porphyry copper deposit. In the method, the multiscale feature extractor is integrated into the encoder to fully exploit and mine the latent feature representations of hyperspectral data, and the decoder reconstructs the original inputs with the latent and class vectors. There are some problems that need to be further considered. The specific opinions are as follows:

1.      In your multi-scale feature extractor, how do you know the way you fuse multiscale feature is proper, you could give more comparison experiments.

2.      The data in your article is hyperspectral data, but the method you propose dose not take full advantage of the spectral information.

3.      In your experiment part, you just compare one Semi-supervised method SXGBoost with SSAAE, could there be more method to compare?

4.      In the experiment with Drilling core Hyperspectral Dataset, the overall sample number is too small, making the result unconvincing.

Author Response

Dear Reviewer:

Thanks for your valuable comments, which can better improve the quality of this manuscript. 
Please see the attachment, where we provided a point-by-point response to your comments 

With best Regards,

Xu Yang, Jianguo Chen, Zhijun Chen

China University of Geosciences (Wuhan)

Reviewer 2 Report

Hello to dear authors

The conducted research is an innovative and practical study. This manuscript has made good use of methods based on artificial intelligence and hyperspectral data in the analysis of satellite images in order to separate alteration zones. The results of this research can help identify alteration zones in similar areas. There are points of view to improve the presentation of research in a standard way, which are mentioned below.

Abstract:

(1) Line 12 : Please use capital letter for  Semi-Supervised Adversarial Autoencoder (SSAAE). What does the "E" stand for? . It is better to write "In the present study, a Semi-Supervised Adversarial Autoencoder (SSAAE) ) was proposed to classify the alteration zones based on hyperspectral data of drill cores from the Pulang porphyry copper deposit.

Key words:

(2) Line 27: Please rewrite all keywords, you should not use the same words as the title.

Introduction:

(3) Line 51 : Please write the abbreviation for the spectral feature fitting.

(4) In order to make the presentation of the article more transparent, please mention the research objectives in a numbered form at the end of this section.

(5) According to the claim announced in this research, the hyperspectral data obtained from the drilled cores in the deposit were used in the analysis of satellite images. As you know, surface alterations that can be identified in the form of halos are different from underground alterations (due to the effect of surface factors). How do you justify this difference? Please explain this more clearly in the introduction section.

Data Collection Method and Preprocessing:

(6) Due to the fact that in this part, the extraction of hyperspectral data and their pre-processing from Polang deposit is explained, it should be placed in the "Results and Analysis" section.

(7) I suggest that the title of this sub-section be changed to "Hyperspectral Data Extraction from Drilled Cores".

(8) Line 128 : It is more professional to use the passive structure in the writing of the manuscript, because everyone knows that you have done this research and there is no need to emphasize the subject "We" throughout the text.

Proposed Method:

(9) The title of this section is not standard. Please choose a proper title such as "Methodology" or "Technical Follow".

(10) This section is a little large and boring. In order to improve the presentation of the methodology, please prepare a flowchart and place it in this section. This flowchart should present the general methodology (and innovation) used in the research from the beginning (data input) to the end (result output) in a simple way.

(11) Line 164 : AE's full name has not been mentioned before. In this section, mention it completely.

Experiments and Analysis:

(12) Please change the title to "Results and Analysis".

(13) What is your validation for identified alteration zones based on SSAAE? If possible, use a validation method, otherwise explain this issue more clearly.

References:

As stated in the literature review, recently, in order to separate the alteration zones, a lot of research has been done on machine learning and artificial intelligence in the analysis of satellite images. Some of the references used are outdated. Please use the following researches in your manuscript:

Alteration Detections Using ASTER Remote Sensing data and Fractal Geometry for Mineral prospecting in Hemich Area, NE Iran

Neuro-Fuzzy-AHP (NFAHP) Technique for Copper Exploration Using Advanced Spaceborne Thermal Emission and Reflection Radiometer (ASTER) and Geological Datasets in the Sahlabad Mining Area, East Iran

Fusion of Remote Sensing, Magnetometric, and Geological Data to Identify Polymetallic Mineral Potential Zones in Chakchak Region, Yazd, Iran

Multi-Dimensional Data Fusion for Mineral Prospectivity Mapping (MPM) Using Fuzzy-AHP Decision-Making Method, Kodegan-Basiran Region, East Iran

I hope that your manuscript will be ready for publication after the general corrections have been revised carefully.

Best regards

Author Response

Dear Reviewer:

Thanks for your valuable comments, which can better improve the quality of this manuscript. 
Please see the attachment, where we provided a point-by-point response to your comments.

With Best Reagrds,

Xu Yang, Jianguo Chen, Zhijun Chen

China University of Geosciences (Wuhan)
